# Late Relapse after Allogeneic Stem Cell Transplantation in Patients Treated for Acute Myeloid Leukemia: Relapse Incidence, Characteristics, Role of Conditioning Regimen, and Outcome

**DOI:** 10.3390/cancers16071419

**Published:** 2024-04-05

**Authors:** Chloé Antier, Maxime Jullien, Benoît Tessoulin, Marion Loirat, Pierre Peterlin, Alice Garnier, Amandine Le Bourgeois, Patrice Chevallier, Thierry Guillaume

**Affiliations:** Department of Hematology, Nantes University Hospital, Hôtel-Dieu, F-44903 Nantes, France; chloe.antier@chu-nantes.fr (C.A.); maxime.jullien@chu-nantes.fr (M.J.); benoit.tessoulin@chu-nantes.fr (B.T.); m.loirat@ch-saintnazaire.fr (M.L.); pierre.peterlin@chu-nantes.fr (P.P.); alice.garnier@chu-nantes.fr (A.G.); amandine.lebourgeois@chu-nantes.fr (A.L.B.); patrice.chevallier@chu-nantes.fr (P.C.)

**Keywords:** allogeneic hematopoietic stem cell transplantation, relapse of acute myeloid leukemia, late relapse following alloHSCT

## Abstract

**Simple Summary:**

Relapse following allogeneic hematopoietic stem cell transplantation (alloHSCT) for acute myeloid leukemia (AML) is the main reason for treatment failure. Most relapses occur during the first six months. We have observed in recent years that some patients relapse late, beyond 2 years after allograft. We sought to evaluate the frequency and the risk factors associated with these late relapses. We observed that these late relapses affect a significant number of patients, that the absence of chronic GvHD is more often associated. In addition, the intensity of the conditioning regimen does not seem to play a role and it is possible to re-treat successfully these patients. We conclude that prolonged monitoring after alloHSCT for AML is recommended.

**Abstract:**

Late relapse, beyond 2 years following alloHSCT for AML, is rare. Among the 376 patients allografted for AML in our center between 1990 and 2016, 142 (38%) relapsed. The majority (68%) of relapses occurred during the first year following transplantation. Beyond 2 years after alloHSCT, relapse was observed in 26 patients, representing 6.9% of the whole transplanted cohort and 18.3% of the relapsing patients. Cytogenetics at relapse was available in 21 patients and remained for 15 of them concordant to that at diagnosis. The majority (85.7%) of the patients were in CR prior to transplant. Thirteen patients had grade 1–2 acute GvHD, while 13 other patients had grade 3–4 acute GvHD. None of these patients subsequently developed chronic GvHD. In multivariate analyses, a predictive factor of the absence of relapse 2 years after transplantation was the development of extensive chronic GVHD. Salvage therapy achieved new CR in 77% of these patients. We conclude that late relapse can affect a significant minority of patients allografted for AML, and the intensity of the conditioning regimen does not seem to have an impact on these relapses. Moreover, we were able to show that those patients can receive effective salvage therapy.

## 1. Introduction

Allogeneic hematopoietic stem cell transplantation (alloHSCT) still constitutes the main potential curative treatment strategy for patients with acute myeloid leukemia (AML). Unfortunately, relapse remains the main reason for treatment failure of this therapeutic approach, with an incidence of recurrence of about 40 to 50% [1,2]. Most relapses occur during the first six months [3,4]. However, relapses have also been observed beyond 2 years [3,5]. Factors associated with late relapses are not well determined. Disease biology of AML, namely, cytogenetics abnormality, is well known to affect the risk of relapse following alloHSCT. We reviewed early relapses compared to late relapses in a single center in order to identify possible factors associated with late relapses. In our institution, reduced-intensity conditioning (RIC) regimens have been used since the early 2000s, and their use has increased rapidly. The possible role of the intensity of the conditioning in late relapse as well as the clinical presentation and the presence of GvHD have been particularly studied in patients presenting late relapse.

## 2. Patients and Methods

### 2.1. Patient Selection

For this single-institution retrospective study, we reviewed a cohort of patients with the World Health Organization-defined diagnosis of AML [6] who received a first alloSCT in our center between 1 January 1990 and 31 December 2016. The AML prognosis was classified according to ELN 2017 when cytogenetics and/or molecular analysis was available [7]. Biphenotypic leukemias were excluded. Individual data for each patient were collected from the EBMT Promise registry as well as from the electronic patient records concerning possible risk factors that could influence outcome, relapse, and survival. Clinical charts were directly reviewed to update the follow-up. Three hundred seventy-six patients with AML were transplanted during this period. The donors were HLA-identical sibling donors (matched sibling donors (MSDs) with 10 out of 10 matched HLA-A, B, C, DRB1, and DQB1 alleles), 9 out of 10 mismatched unrelated or matched unrelated volunteer donors (MUDs), cord blood donors, or haplo-identical donors.

### 2.2. Treatment Characteristics

The patients underwent conditioning using various preparative regimens, and these were classified as myeloablative conditioning (MAC), reduced-intensity conditioning (RIC), or non-myeloablative (NMA) according to the accepted criteria [8,9]. RIC/NMA consisted of 150 mg/m^2^ fludarabine, 6.4 to 9.6 mg/kg busulfan, 5 mg/kg antithymocyte globulin (ATG) or 90 mg/m^2^ fludarabine, and 2-Gy total body irradiation (TBI) for MUD or MRD, and 50 mg/kg cyclophosphamide, 200/mg/m^2^ fludarabine, and 2-Gy TBI for cord blood transplant. MAC consisted of 120 mg/kg cyclophosphamide and 6.4 mg/kg busulfan or 100 mg/kg cyclophosphamide and 12-Gy TBI or 150 mg/m^2^ fludarabine, 12.8 mg/kg busulfan, and 5 mg/kg antithymocyte globulin (ATG). The haplo-identical conditioning regimen consisted of 150 mg/m^2^ fludarabine or 150 mg/m^2^ clofarabine, 29 mg/kg cyclophosphamide, and 2-Gy TBI, followed by 100 mg/kg post-transplant cyclophosphamide. Sequential conditioning consisted of 4 g/m^2^ cytarabine, 120 mg/m^2^ fludarabine or 150 mg/m^2^ clofarabine, 60 mg/kg cyclophosphamide, 6.4 mg/kg busulfan, and 5 mg/kg ATG. Institutional GVHD prophylaxis consisted of cyclosporine with methotrexate or mycophenolate mofetil based on the conditioning regimen and type of donor.

### 2.3. Statistical Analysis

Statistical analyses were performed using R software version 4.1.3. The probability of overall survival was calculated using Kaplan–Meier estimates, and the log-rank test was used for univariate comparisons for all variables assessed. Cumulative incidence rates of relapse and transplant-related mortality were estimated using Fine and Gray competing risk regression. The *p*-value was set at ≤0.05 for statistical significance. Population characteristics were compared using the X^2^ test for discrete variables and Student’s *t*-test for continuous variables. Multivariate analyses were performed using the Cox proportional hazard model.

## 3. Results

### 3.1. Patient Characteristics

Between 1 January 1990 and 31 December 2016, a total of 376 patients with AML were transplanted. Their characteristics are summarized in Table 1. The median age at transplant was 48.8 years old (18.3–71.0). The majority was male (51.6%). Cytogenetics at diagnosis was favorable in 10.9% of the patients, intermediate in 42.3%, and unfavorable in 27.9%. For 18.9% of the patients, cytogenetics was either not completed (mainly for patients treated in the early 1990s) or missing.

The median time between diagnosis and transplantation was 4.5 months. At the time of grafting, 81% of the patients were in complete remission (CR) and 19% had active disease (defined as the absence of complete remission, including incomplete count recovery). Donor types included MSD (47%), MUD (33%), cord blood (10%), mismatched unrelated 9/10 (6%), and haplo-identical donors (4%). In this study, 32% of the patients received a MAC regimen.

### 3.2. Outcome and Factors Affecting Survival, Relapse, and TRM

This cohort of patients had a median follow-up of 8.6 years (103.6 months; CI 95% 71.6–159.5 months), with a 5-year overall survival (OS), relapse-free survival (RFS), and transplant-related mortality (TRM) of 50.3% (CI 95% 45.5–55.6%), 46.9% (CI 95% 42.1–52.2%), and 17% (CI 95% 13–21%), respectively. The relapse rate at 2 years and 5 years was 31% and 36% (CI 95% 26–36% and 31–41%), respectively.

In univariate analysis, several factors had a negative impact on OS: Unfavorable ELN 2017 subgroup (HR 2.06, *p* = 0.01), active disease at the time of transplant (HR 3.67, *p* < 0.001), and sequential conditioning (HR 2.80, *p* < 0.001) were unsurprisingly associated with a lower survival rate (Table 2). In multivariate analysis, active disease at transplant was the sole adverse factor (HR 3.7375, *p* < 0.001) (Table 2). A similar conclusion was drawn regarding analysis of RFS. Active disease at transplant was the only factor associated with relapse in a multivariate analysis (HR 3.06, *p* = 0.001, Table 3). Regarding TRM, in multivariate analysis, female gender had a favorable impact (HR 0.47, *p* = 0.026), while active disease at transplant was associated with unfavorable outcome (HR 2.05, *p* = 0.033).

### 3.3. Factors Associated with Late Relapse

Among the 376 patients, 142 (37.8%) relapsed (Figure 1). The majority (67.6%) of relapses occurred during the first year following transplantation. Table 4 summarizes the incidence of relapse according to the time in years following allotransplant. Beyond 2 years after transplantation, relapse was observed in 26 patients, representing 6.9% of the whole transplanted cohort and 18.3% of the relapsing patients. In this group of late relapses, the median age was 53.9 years, with the majority being women (65.4%). These 26 patients were divided according to the ELN 2017 classification (at diagnosis) into favorable, intermediate, and unfavorable subgroups of 7.7%, 38.5%, and 34.6%, respectively, and 19.2% could not be classified. Twenty-one patients (80.1%) presented a medullary relapse, while 4 patients had only an extramedullary relapse (4 granulocytic sarcoma), 2 relapsed as myelodysplastic syndrome, and 1 had a molecular relapse. Among those with a medullary relapse, one had a cutaneous granulocytic sarcoma and another had neuromeningeal involvement.

The majority (84.6%) of the late-relapse patients were in CR prior to transplant. RIC was used in 61.5% of the patients, MAC in 30.8%, and sequential in 7.7%. Stem cell sources were peripheral blood stem cells in 65.4%, bone marrow in 23.1%, and cord blood in 11.5%. Twelve patients had grade 1–2 acute GvHD, while three other patients had grade 3–4 acute GvHD. One patient developed extensive chronic GvHD.

For the analysis of factors associated with late relapse, we focused on the patients who were alive without relapse 2 years after transplant (n = 199) and compared patients who relapsed beyond two years after transplant (late-relapse group) with those who did not relapse beyond two years (no-relapse group) (Table 5). We found that the incidence of chronic GVHD was significantly different between the two groups, with a quarter of patients without relapse presenting chronic GVHD and only 1 of the 26 late relapses having chronic GVHD (*p* = 0.03).

### 3.4. Survival of Patients beyond 2 Years from Transplantation

For patients surviving at least 2 years after transplantation, the probability of survival at 1 year, 2 years, and 5 years was 97.5%, 94.9%, and 88.4%, respectively (Figure 2), while the disease-free survival was 91.9%, 89.3%, and 86.2%, respectively. For the same group of patients, the relapse rate incidence at 1 year, 2 years, and 5 years was 6.6%, 8.7%, and 11%, respectively.

### 3.5. Treatment of Late AML Relapse

All the patients presenting late relapse were treated (Figure 1). The majority of patients (20 out of 26, 77%) obtained a new CR following salvage treatment. Ten (38%) patients received new intensive chemotherapy, allowing for a second CR. Among these 10 patients, 9 received a second allogeneic HSCT. Following this second transplantation, unfortunately, six patients relapsed and died due to relapse, two patients died from transplant-related mortality (TRM), and only two patients remained alive and in remission.

Eleven other patients received non-intensive azacitidine-based chemotherapy, allowing for a new CR for nine patients. Only two of these patients received a second alloHSCT: One of them is alive, while the other died from TRM. Azacitidine was associated with donor lymphocyte infusion (DLI) in two patients; one of these two patients is alive, while the other died from relapse. Azacitidine was combined with radiotherapy in two patients with myeloid sarcoma. Unfortunately, these two patients died from relapse. Finally, five patients received azacitidine alone, and all of them but one had died by the last follow-up (three from relapse and one from TRM).

The last five patients received a form of treatment other than chemotherapy: One patient received DLI, a second patient received radiotherapy alone, and a third patient received DLI and radiotherapy, all without efficiency. A fourth patient was included in a clinical trial. All four of these patients died from relapse. A fifth patient presented a molecular relapse and was treated with imatinib.

The median overall survival of the patients presenting late relapse was 28 months and remains far superior to the survival of patients with more precocious relapses (Figure 3).

### 3.6. Cytogenetic Characteristics at Late Relapse

In 21 patients, a karyotype was obtained at relapse and compared to that at diagnosis. Fifteen patients had concordant karyotype at diagnosis and relapse, while six patients had new cytogenetic abnormalities (two monosomy 7, one complex caryotype, one t(1;16), one add(11q), and one t(X;20)). A molecular analysis was performed on only eight patients at relapse: Identical mutations were observed in six patients, Flt3 mutation newly appeared in one patient, and another patient had a loss of the Flt3 mutation.

### 3.7. Chimerism at Relapse

CD3+ and/or CD34+ cell chimerisms were obtained at relapse in seven and five patients, respectively. CD34+ cell chimerism was always mixed, but two patients had a complete CD3+ cell chimerism at relapse. The first one had a CD3+ cell chimerism at 95% and received an intensive salvage treatment, followed by a second alloHSCT. The other one had a CD3+ cell chimerism at 100% and was only a molecular relapse. No CD34+ cell chimerism was available for those two patients.

## 4. Discussion

AlloSCT is proposed as an effective consolidation therapy and possible curative option in patients with AML. Unfortunately, the incidence of relapse remains quite high and is estimated to be around 40 to 50%, depending on the aggressiveness of the disease. Although most relapses occur during the first 6 months, the incidence of relapses occurring beyond one year remains significant and is estimated to be around 10% [3,4]. We observed that a few patients relapsed several years following alloSCT, and therefore, we reviewed late relapses of AML in our center to find out whether we could identify factors that might be associated with these late relapses, particularly whether the intensity of the conditioning regimen might play a role.

The first step was to evaluate the incidence of early and late relapse according to the time period. Late relapse was defined as relapse occurring beyond 2 years following alloSCT. This time threshold was chosen because the large majority (82% [3]) of post-transplant AML relapses occurs within 2 years after grafting. In our retrospective analysis, we observed that 38% of the allografted patients relapsed. These relapses occurred, for 80.5% of them, during the first 2 years. This observation is comparable to previous reports [3,4]. Relapse could be observed very late up to 8 years. Late “relapsers” represented 7.4% of allografted patients.

The probability for survival at 5 years after alloSCT for patients alive at 2 years was 89%. This is comparable to a previous report where the probabilities of survival and disease-free survival at 10 years, for those in remission and alive at 2 years, were 84% and 82%, respectively [10], with a cumulative incidence of relapse of AML at 10 years estimated to be 10%.

RIC leads to lower treatment-related mortality (TRM) but with a higher risk of relapse compared to MAC, as reported in retrospective trials [11,12,13,14,15]. Three randomized studies have compared RIC to MAC in patients allotransplanted for AML and myelodysplastic syndrome (MDS) [16,17,18]. All these trials were stopped before completion due to poor accrual or at the request of the safety-monitoring board. Two studies did not find differences in terms of OS and relapse [17,18]. The third and more recent study, with a median follow-up of 4 years [16], showed that TRM increased over time in patients who received MAC; however, this increase was not sufficient to overcome the significantly higher earlier relapse rates seen in patients who received RIC, and therefore, OS remained significantly better for patients receiving MAC. Examining our group of late relapsers, we did not find that the intensity of the conditioning regimen had an impact on these late relapses. Chronic GvHD appears to be a protective factor for late relapse. Wingard et al. found that for AML, later stages of disease before transplantation, absence of acute GVHD, lower performance scores at the time of transplantation, and T-cell depletion of the hematopoietic graft were associated with higher likelihood for late relapse.

Extramedullary relapses have been observed in a fifth of late-relapsing patients, as has previously been reported [19]. Frequently, the median time to extramedullary relapse is longer than that to medullary only [19,20,21,22]. Several predictive factors of extramedullary relapses have been reported: a previous extra-medullary presentation, French–American–British classification M4/M5 leukemia, high-risk cytogenetics, and advanced disease status at the time of transplantation [23].

The reasons why leukemic cells remain dormant for several years after allografting and suddenly are the source of a frank relapse is not well understood. Different hypotheses have been proposed: a clonal evolution of the initial clone and/or various mechanisms of immune escape. Ding et al. found that the founding clone or a subclone of the founding clone gained mutations and expanded at relapse [24]. This founder leukemic clone seems to be most frequently responsible for late relapse [25]. In our cohort, the comparison of the cytogenetics between the initial diagnosis and the relapse was performed in 20 out of our 28 late-relapsing patients. We observed that only 25% had a modification of their karyotype. In addition, only six patients had molecular analysis at relapse. It may be likely that the initial clone was responsible for the late relapse. However, the low number of patients and the lack of complete molecular analysis at relapse may not confirm this hypothesis.

In addition, the interface between T cells and leukemic cells may change significantly following alloHSCT. Loss of costimulatory molecules in T cells and altered expression of inhibitory molecules in T and leukemic cells has been involved in relapse [26,27]. Furthermore, genomic loss of a mismatched HLA haplotype, as well as transcriptional silencing of HLA molecules, may occur in relapsing leukemia.

As classically reported, rescue treatment for late relapse is much more efficient than treatment of early relapse. The majority (75%) of our patients obtained a new remission, and half of our late relapsers were alive 2 years following rescue treatment. There are not specific recommendations regarding the choice of treatment: Some patients may be able to support new intensive chemotherapy, but the majority will receive hypomethylating agents. The addition of immunotherapy with DLI or a second transplantation might further improve remission and prolong survival.

## 5. Conclusions

Late relapse of AML in allografted patients may occur in a significant group of patients. For those patients for whom cytogenetics of leukemic cells at relapse was available, the majority remained concordant to that at diagnosis. Intensive salvage treatment appeared to be effective in our series, and a second allo transplant might prolong survival.

## Figures and Tables

**Figure 1 cancers-16-01419-f001:**
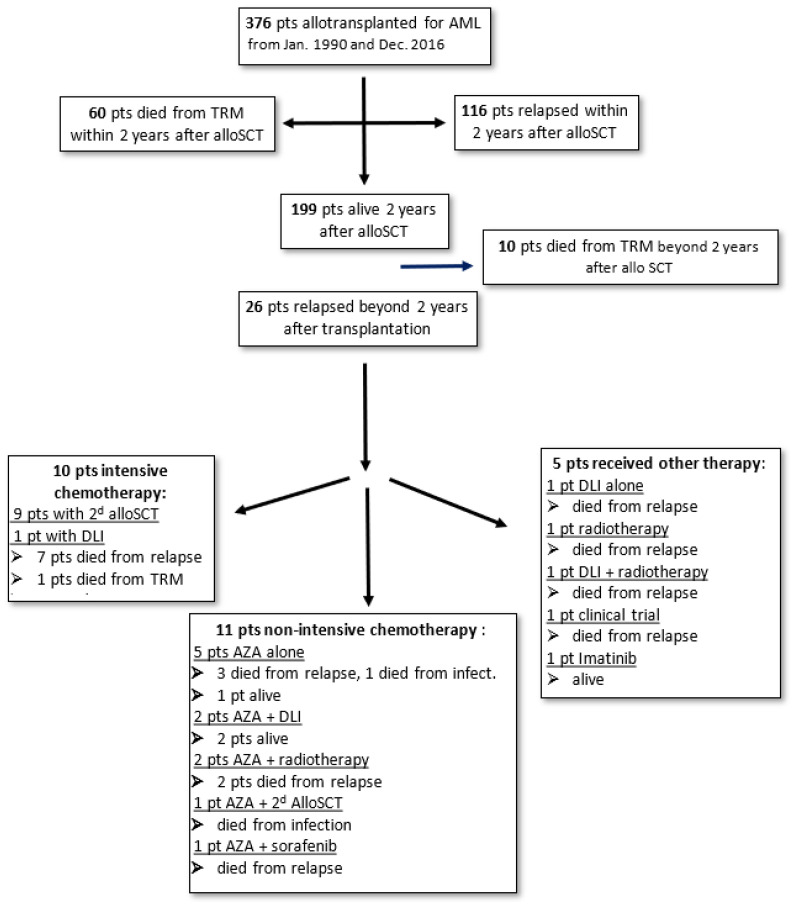
Diagram summarizing the outcome of the patients and treatment received.

**Figure 2 cancers-16-01419-f002:**
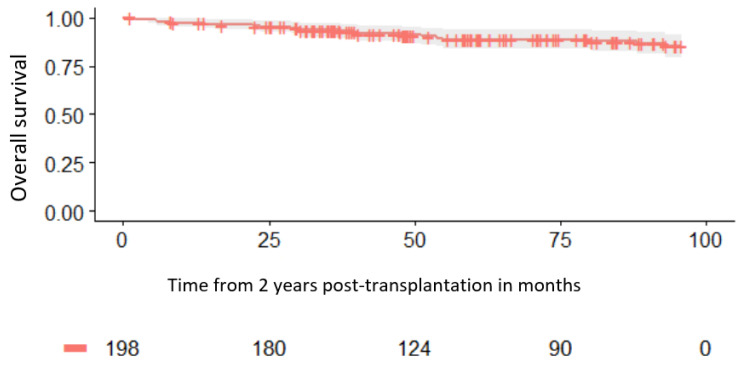
Survival for patients alive at 2 years.

**Figure 3 cancers-16-01419-f003:**
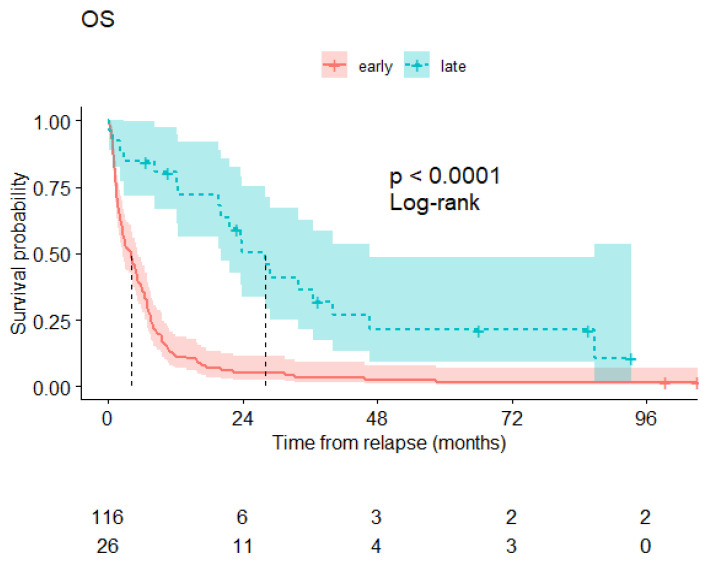
Overall survival of relapsing patients from the time of relapse diagnosis: patients relapsing before 2 years after transplantation (red curve) and those relapsing beyond 2 years (blue curve).

**Table 1 cancers-16-01419-t001:** Characteristics of the whole group of transplanted patients.

	*n* = 376	%
Gender		
Male	194	51.6%
Female	182	48.4%
Median age (range)	48.8 y (18.4–71.0)	
ELN classification		
Favorable	41	10.9%
Intermediate	159	42.3%
Unfavorable	105	27.9%
NA	71	18.9%
Status at grafting		
CR	304	80.9%
Refractory	72	19.1%
Type of donor		
Identical sibling	177	47.1%
Matched unrelated	124	33%
Haplo-identical	16	4.3%
Mismatched unrelated 9/10	23	6.1%
Cord blood	36	9.6%
Type of graft		
PBSC	244	64.9%
BM	96	25.5%
CB	36	9.6%
Conditioning		
MAC	120	31.9%
RIC	209	55.6%
SEQ	47	12.5%
Year of transplantation		
Before 2005	113	30.1%
After 2005	263	69.9%

NA, not available; CR, complete remission; MAC, myelo-ablative conditioning; RIC, reduced-intensity conditioning.

**Table 2 cancers-16-01419-t002:** Univariate and multivariate analysis of potential risk factors impacting overall survival.

Characteristics	Variable (Percentage)	HR (Univariate)	HR (Multivariate)
Median age at transplant	48.8 years (18.3–71.0)	1.01 (1.00–1.02, *p* = 0.26)	-
Gender			
Male	194 (51.6%)	-	-
Female	182 (48.4%)	0.68 (0.51–0.90, *p* = 0.008)	0.89 (0.65–1.23, *p* = 0.48)
ELN 2017 subgroup			
Favorable	41 (10.9%)	-	-
Intermediate	159 (42.3%)	1.26 (0.73–2.16, *p* = 0.40)	1.14 (0.66–1.95, *p* = 0.65)
Adverse	105 (27.9%)	2.06 (1.20–3.55, *p* = 0.009)	1.50 (0.85–2.64, *p* = 0.16)
Status at transplant			
CR	304 (80.9%)	-	-
Refractory	72 (19.1%)	3.67 (2.71–4.99, *p* < 0.001)	3.75 (1.95–7.22, *p* < 0.001)
Conditioning			
RIC	209 (55.6%)	-	-
MAC	120 (31.9%)	1.00 (0.72–1.38, *p* = 0.99)	0.82 (0.53–1.26, *p* = 0.36)
Sequential	47 (12.5%)	2.80 (1.93–4.06, *p* < 0.001	0.73 (0.37–1.46, *p* = 0.37)
Type of graft			
PBSC	244 (64.9%)	-	-
BM	96 (25.5%)	0.87 (0.62–1.22, *p* = 0.43)	-
CB	36 (9.6%)	1.06 (0.67–1.68, *p* = 0.81)	-
Type de donor			
MSD	177 (47.1%)	-	-
MUD	124 (33%)	1.15 (0.84–1.58, *p* = 0.39)	-
Haplo	16 (4.3%)	1.49 (0.75–2.97, *p* = 0.25)	-
9/10	23 (6.1%)	1.26 (0.71–2.27, *p* = 0.43)	-
CB	36 (9.6%)	1.20 (0.74–1.93, *p* = 0.46)	-

**Table 3 cancers-16-01419-t003:** Univariate and multivariate analyses of potential risk factors impacting relapse-free survival.

Characteristics	Variable (Percentage)	HR (Univariate)	HR (Multivariate)
Median age at transplant	48.8 years (18.3–71.0)	1.01 (1.00–1.02, *p* = 0.22)	-
Gender			
Male	194 (51.6%)	-	-
Female	182 (48.4%)	0.70 (0.53–0.92, *p* = 0.01)	0.90 (0.66–1.23, *p* = 0.52)
ELN 2017 subgroup			
Favorable	41 (10.9%)	-	-
Intermediate	159 (42.3%)	1.29 (0.75–2.20, *p* = 0.35))	1.17 (0.68–2.00, *p* = 0.58))
Adverse	105 (27.9%)	2.24 (1.30–3.85, *p* = 0.004)	1.66 (0.95–2.91, *p* = 0.08)
Status at transplant			
CR	304 (80.9%)	-	-
Refractory	72 (19.1%)	3.66 (2.70–4.94, *p* < 0.001)	3.06 (1.59–5.89, *p* = 0.001)
Conditioning			
RIC	209 (55.6%)	-	-
MAC	120 (31.9%)	0.90 (0.65–1.23, *p* = 0.51)	0.83 (0.54–1.26, *p* = 0.37)
Sequential	47 (12.5%)	2.95 (2.04–4.26, *p* < 0.001)	0.90 (0.45–1.80, *p* = 0.76)
Type of graft			
PBSC	244 (64.9%)	-	-
BM	96 (25.5%)	0.80 (0.58–1.11, *p* = 0.19)	-
CB	36 (9.6%)	1.11 (0.71–1.74, *p* = 0.65)	-
Type de donor			
MSD	177 (47.1%)	-	-
MUD	124 (33%)	1.17 (0.86–1.60, *p* = 0.32)	-
Haplo	16 (4.3%)	1.34 (0.68–2.67, *p* = 0.40)	-
9/10	23 (6.1%)	1.43 (0.83–2.47, *p* = 0.20)	-
CB	36 (9.6%)	1.30 (0.82–2.08, *p* = 0.26)	-

**Table 4 cancers-16-01419-t004:** Successive timing of relapses following allogeneic HSCT.

Period of Time Post-Transplant	Relapses *N* = 142 (37.7%)	Patients Alive without Relapse at the End of Each Post-Transplant Year
First year (months 0 to 12)	97 relapses/376 (26%)	222
Second year (months 13 to 24)	19 relapses/221 (8.1%)	199
Third year (months 25 to 36)	13 relapses/198 (7%)	179
Fourth year (months 37–48)	4 relapses/179 (2.2%)	170
Fifth year (months 49–60)	2 relapses/170 (1.7%)	147
Sixth year (months 61–72)	1 relapses/146 (1.3%)	*
Seventh year (months 73–84)	1 relapse/	*
Eighth year (months 85–96)	no relapse/	*
Ninth year (months 97–108)	3 relapses/	*
Tenth year (months 108–120)	2 relapses/	*

* The follow-up was not long enough for all patients to be able to provide a relative value.

**Table 5 cancers-16-01419-t005:** Characteristics of patients presenting a relapse beyond 2 years after transplantation compared to those of patients alive and in CR two years after transplantation.

Characteristics	Late Relapse (*n* = 26)	No Relapse (*n* = 170)	*p*-Value (Chi^2^ or *t*-Test)
Median age at transplant	53.9	47.7	0.046
Sex			
Male	9 (34.6%)	79 (45.7%)	0.398
Female	17 (65.4%)	94 (54.3%)	
ELN 2017 subgroup			
Favorable	2 (7.7%)	25 (14.5%)	0.265
Intermediate	10 (38.5%)	81 (46.8%)	
Adverse	9 (34.6%)	32 (18.5%)	
NA	5 (19.2%)	35 (20.2%)	
Status at transplant			
CR	22 (84.6%)	163 (94.2%)	0.169
Active disease	4 (15.4%)	10 (5.8%)	
Conditioning			
RIC	16 (61.5%)	103 (59.5%)	0.739
MAC	8 (30.8%)	62 (35.8%)	
Sequential	2 (7.7%)	8 (4.6%)	
Graft source			
PBSC	17 (65.4%)	108 (62.4%)	0.827
BM	6 (23.1%)	49 (28.3%)	
CB	3 (11.5%)	16 (9.2%)	
Type of donor			
MSD	11 (42.3%)	86 (49.7%)	0.516
MUD	9 (34.6%)	57 (32.9%)	
Haplo	0 (0%)	6 (3.5%)	
09/10	3 (11.5%)	8 (4.6%)	
Cord blood	3 (11.5%)	16 (9.2%)	
Acute GVHD			
No	11 (42.3%)	85 (49.1%)	0.801
Grade 1–2	12 (46.2%)	69 (39.9%)	
Grade 3–4	3 (11.5%)	19 (11%)	
Chronic GVHD			
None	25 (96.2%)	129 (74.6%)	0.028
Extensive	1 (3.8%)	44 (25.4%)	

## Data Availability

Data used in this work are available upon reasonable request from the corresponding author.

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
