# Peer review of "Late Relapse after Allogeneic Stem Cell Transplantation in Patients Treated for Acute Myeloid Leukemia: Relapse Incidence, Characteristics, Role of Conditioning Regimen, and Outcome"

_cancers, 2024, doi:10.3390/cancers16071419_

Round 1

Reviewer 1 Report

Comments and Suggestions for Authors

I recommend improving the introduction of this study. 

The second advice is to introduce some figures that illustrate the results. There are many tables and is very hard to read this article

Author Response

Reviewer 1

I recommend improving the introduction of this study.

We thank the reviewer for his advice. We modified the text, adding several sentences that might better explain the purpose of the study, namely the attempt to identify factors associated with late relapse.

The second advice is to introduce some figures that illustrate the results. There are many tables and is very hard to read this article

Again, thank you for this pertinent advice. We add a figure showing the comparison of the OS for early and late relapses, and survivals according to genetic abnormalities. Figure 1 was modified with more details to be clearer.

Reviewer 2 Report

Comments and Suggestions for Authors

This is an interesting paper concerning late AML relapses after HSCT in adults.

However some issues should be improved:

1. There is no information about time of the follow-up after late relapse.

2.There is no survival curves for the late relapse group.

3. Conclusions contain information that half of the patients with late relapse were alive 2 years following rescue treatment (line 255) while there is no such an information in the results. Moreover, it is stated in the results that only 18% of the patients with late relapse remained in CR at the time of analysis (line 171-172) and 19 patients  died from relapse (line 173). Figure 1 suggests that 16 pts died and there is no information about outcome of 5 pts who received other therapy. It should be clarified.

Reviewer 3 Report

Comments and Suggestions for Authors

In the manuscript of “Late relapse after allogeneic stem cell transplantation in patients treated for acute myeloid leukemia: Relapse incidence, characteristics, role of conditioning regimen, and outcome.”, the authors reviewed early relapses compared to late relapses in a single center. It was confirmed that extensive chronic GVHD was a predictive factor of the absence of relapse 2 years after transplantation. And the conditioning regimen did not impact patients’ relapses. However, this paper is unacceptable for publication in its current form. The specific comments are listed below:

Q1. There exists 49 pts not mentioned of their conditionings in Table 1. Please give the explanation.

Q2. Data in tables should be values with decimal places, not integer values.

Q3. The subtitle “Outcome, OS and PFS” seems not suitable.

Q4. Please indicate the header of Table 2 and 3, when describing univariate and multivariate analyses. And the format should be altered.

Q5. Please add axis titles of Figure 2.

Q6. I wonder if the status at transplant, conditioning regimen, source du griffon or type of donor could affect the OS or PFS of patients in this cohort. It would be better if authors could analyze the K-M curves.

Q7. What about the genetic feature at diagnosis of these AML patients? What’s more, authors should discuss the functions of basic genetic status on outcomes of these patients.

Round 2

Reviewer 1 Report

Comments and Suggestions for Authors

NA

Reviewer 2 Report

Comments and Suggestions for Authors

It can been accepted in present form

Reviewer 3 Report

Comments and Suggestions for Authors

In the manuscript of “Late relapse after allogeneic stem cell transplantation in patients treated for acute myeloid leukemia: Relapse incidence, characteristics, role of conditioning regimen, and outcome.”, the authors reviewed early relapses compared to late relapses in a single center. It was confirmed that extensive chronic GVHD was a predictive factor of the absence of relapse 2 years after transplantation. And the conditioning regimen did not impact patients’ relapses. What’s more, this letter can be accepted for publication.